

# Influence of geographic origin and tissue type on the medicinal chemical compounds of *Semiliquidambar cathayensis*

Xiaoming Tian[1], Hao Lv[1,2], Guangfeng Xiang[1], Jing Peng[1], Gaofei Li[1], Yan He[1], Fan Zhang[1] and Cun Mou[1,2]

[1] Institute of Plant Conservation, Hunan Botanical Garden, Changsha, China
[2] Faculty of Life Science and Technology, Central South University of Forestry and Technology, Changsha, China

## ABSTRACT

*Semiliquidambar cathayensis* is widely used in traditional Chinese medicine owing its high concentrations of polyphenol, triterpenoidic acid, and flavonoids. This study aimed to explore the impact of geographical origin and tissue type on the contents of chemical compounds of *S. cathayensis*, as determined by colorimetric and chromatographic methods. Therefore, we quantitively evaluated chemical compounds found in the tissues of various organs of plants collected in six different regions. Overall, we found that geographical origin affected the content of medicinal compounds in *S. cathayensis* leaves, with plants from Jingzhou county showing the best therapeutic potential. However, no specific correlation was observed with latitude. It is noteworthy that the amount of paeoniflorin and other compounds can be used as biomarkers of geographical origin and tissue type. Most medicinal compounds accumulated mainly in the leaves, whereas ursolic and oleanolic acids accumulated in the roots. These results show that the comprehensive medicinal value of the leaves of *S. cathayensis* in Jingzhou county is the highest, but the root should be selected first to collect oleanolic acid and ursolic acid.

## INTRODUCTION

*Semiliquidambar cathayensis*, of the Altingiaceae family, is a traditional medicinal plant with extremely strict habitat requirements, and its natural distribution is gradually decreasing due to the influence of global climate change and human activities. However, because of its high economic value, *S. cathayensis* has been cultivated in many parts of China (*Liu et al., 2022b*). The roots, branches, leaves, bark, and nectar of *S. cathayensis* have high utilization value and have been widely used for a long time in traditional Chinese medicine (*Yang, Liu & He, 2019*). Among the tissues of *S. cathayensis*, the leaves are the most widely used in traditional medicine because they are readily accessible and available (*Tian et al., 2022*; *Yang, Liu & He, 2019*). Pharmacological studies have shown that leaf extracts can relieve pain, modulate inflammation, and alleviate other symptoms; furthermore, they can be

Corresponding authors
Xiaoming Tian,
tianxiaoming1986@126.com
Cun Mou, stevemoon@126.com

used for treating rheumatoid arthritis, postpartum muscle relaxation, and hepatitis, among other conditions (*Liang et al., 2015*; *Wu-Liang et al., 1999*). In particular, *S. cathayensis* is reportedly rich in triterpenes, total flavonoid, and polyphenols (*Yang, Liu & He, 2019*). According to the analysis of *Tian et al. (2018)*, *S. cathayensis* tissues are also rich in various other compounds with pharmacological activities, including oleanolic acid (*Zhou et al., 2002*). Previous studies have shown that *S. cathayensis* can promote blood circulation and remove blood stasis in rats, and inhibit hepatitis B virus (*Sun et al., 2014b*) Currently available data demonstrates that *S. cathayensis* has high medicinal value.

Geographical origin, tissue type, and developmental stage are all factors that affect plant secondary-metabolite contents (*Alessandra et al., 2021*; *Atyane et al., 2017*; *Bat et al., 2018*; *Karsten, Matjaž & Samo, 2021*; *Narae et al., 2021*; *Yuan, Xiao & DaXia, 2021*). In particular, different geographical origins force plants to grow in different habitats to which they must adapt (*Sudha & Ravishankar, 2002*). Due to differences in light, temperature, soil quality, precipitation, and other factors, the production and accumulation of plant secondary metabolites are affected to a certain extent. Thus, for example, *Tilia amurensis* can show a positive response to natural stress, and the precipitation and cultivation environment can affect saffron crops (*Liu et al., 2022a*; *Rashidi et al., 2022*). It is noteworthy that as different plant tissues differ with respect to the secondary metabolites they contain, the tissues of *S. cathayensis* are specifically selected according to their proprieties for use in traditional medicine (*Despina et al., 2022*; *Goufo, Singh & Cortez, 2020*; *Hou et al., 2022*; *Mercado et al., 2021*). Further, different secondary metabolites are synthesized only at specific sites within the plant body and are transported through specific transport pathways, such that, generally, their concentrations differ among different tissues and organs. For example, quinine is only distributed to the bark of *Cinchona* plants (*Wingard, Pecht & Kurz, 1985*). However, the biosynthesis and storage sites of secondary metabolites are not permanent in the plant, and they are often translocated, accumulated, and/or degraded as plant development continues, whereby the contents of medicinal components differ with developmental stage (*Belkheir et al., 2016*). For example, scopolamine alkaloids are mainly synthesized in the roots, but as the plant develops they are translocated to the leaves (*Wingard, Pecht & Kurz, 1985*). In addition, not all secondary metabolites are synthesized at the early stages of development; terpinene, for example, is only synthesized late during development (*Sudha & Ravishankar, 2002*). This evidence shows that there are differences in the composition and content of secondary metabolites in plants at different developmental stages (*De-la Cruz Chacón, Riley-Saldaña & González-Esquinca, 2012*; *Yuan et al., 2016*).

Although the effects of geographic origin and tissue type on soybean, tobacco, and other plant species have been reported, their impact on the content of medicinal components in *S. cathayensis* remains unknown. Further, an in-depth understanding of the influence of these factors on the chemical composition of *S. cathayensis* may help us evaluate the medicinal properties of each part of the plant according to our interests. Therefore, in this study, we determined the contents of total flavonoid, total triterpenoid acid, total polyphenol contents, and other medicinal substances in the different organs of *S. cathayensis* plants collected from different geographical sites.

## MATERIALS & METHODS

### Reagent and standards

All chemicals and solvents used in this study were of analytical grade and were used without any purification. Ursolic and oleanolic acid standards were purchased from Dr. Ehrensorfer (Augsburg, Germany). Chromatographic grade methanol and acetonitrile were purchased from Merck KGaA (Darmstadt, Germany).

### Plant material

*S. cathayensis* has a long medicinal history; its roots, branches, leaves, flowers and fruits have been used in traditional medicine. Oleanolic acid, the active compound of *S. cathayensis*, has inhibitory effect on the antigen of viral hepatitis (*Sun et al., 2014a*), and most of the different flavonoids were enriched in *S. cathayensis* leaves (*Tian et al., 2021*). In October 2021, leaf samples were collected from *S. cathayensis* specimens located at distribution place of six *S. cathayensis* populations with genetic diversity different sites in China (Table 1). At least 100 g pest-free leaves were collected from each geographical origins. Before the test, each leaf was evenly divided into three parts and wiped clean, wrapped in tin foil and frozen in liquid nitrogen for more than 30 min. Samples of the whole plant, including roots, female flowers and male flowers, were collected from an *S. cathayensis* specimen located in Jingzhou, Hunan in April 2021. In October 2021, the roots, stems, leaves and fruits in the same population were collected. After the samples were collected, they were immediately put into an ice box, returned to the laboratory and divided into three portions on average, then frozen with liquid nitrogen for 30 min. Only healthy and disease-free plants were selected for sample collection; flowers were collected during the full-blossom period and fruits were harvested at the best ripening period recommended for consumption (based on internal ethylene concentration and starch index). Three samples of each organ/tissue were collected from each plant as mixed samples (Fig. 1). All plant samples were identified by Lihong Yan and the voucher specimen was deposited in the Herbarium of the Hunan Botanical Garden (http://www.hnfbg.cn/, Lihong Yan, yanlh0424@163.com).

### Determination of total flavonoid, total triterpenoid acid, and total polyphenol by colorimetric assays

The different geographic origin and tissue types of *S. cathayensis* were collected, washed, and dried in shade. The dried samples were powdered and filtered through a 60 mesh sieve. The powder was stored in a sealed bag for subsequent tests. Contents of total flavonoid, total triterpenoid acid, and total polyphenol in different tissues were determined by colorimetric assays.

Total flavonoid content was determined according to the method described by *Zhang et al. (2012)*, with some modifications. The powder (2 g) was placed in 95% ethanol solution (30 mL). The obtained solution was collected by ultrasonic extraction in a water bath at 50 °C for 30 min. The extract procedures were repeated three times and the collected extract solution was blended. Next, the extract solution was passed through the filter paper. The filtrate was cooled for 20 min, then dried with a rotary evaporator. The residue was suspended with methanol (50 mL) and passed through a 0.45 μM membrane (Millipore,

Tian et al. (2023), *PeerJ*, DOI 10.7717/peerj.15484

**Table 1 Geographic distribution of the *Semiliquidambar cathayensis* specimens evaluated.**

| Site | Location | Longitude | Latitude | Altitude (m) | Sample size | Annual average temperature (°C) | Annual average precipitation (mm) | Community type | Canopy density |
|---|---|---|---|---|---|---|---|---|---|
| 1 | Shangsi County, Guangxi Province | 107°50′05″ | 21°40′15″ | 600 | 10 | 22.50 | 2,362.6 | Evergreen broad-leaved forest | 0.80 |
| 2 | Shaxian County, Fujian Province | 117°49′32″ | 26°39′38″ | 835 | 25 | 19.20 | 1,662.4 | Secondary evergreen broad-leaved fores | 0.80 |
| 3 | Wuping County, Fujian Province | 116°17′25″ | 25°21′40″ | 690 | 11 | 19.80 | 1,200 | Coniferous and broad-leaved mixed forest | 0.75 |
| 4 | Dayu County, Jiangxi Province | 114°26′05″ | 25°33′29″ | 730 | 5 | 20.54 | 1,458 | Evergreen broad-leaved forest | 0.75 |
| 5 | Jianghua County, Hunan Province | 111°50′35″ | 24°51′36″ | 625 | 17 | 18.20 | 1,510 | Secondary evergreen broad-leaved forest | 0.70 |
| 6 | Jingzhou County, Hunan Province | 109°30′25″ | 26°32′42″ | 421 | 5 | 16.80 | 1,378 | Secondary evergreen broad-leaved forest | 0.70 |

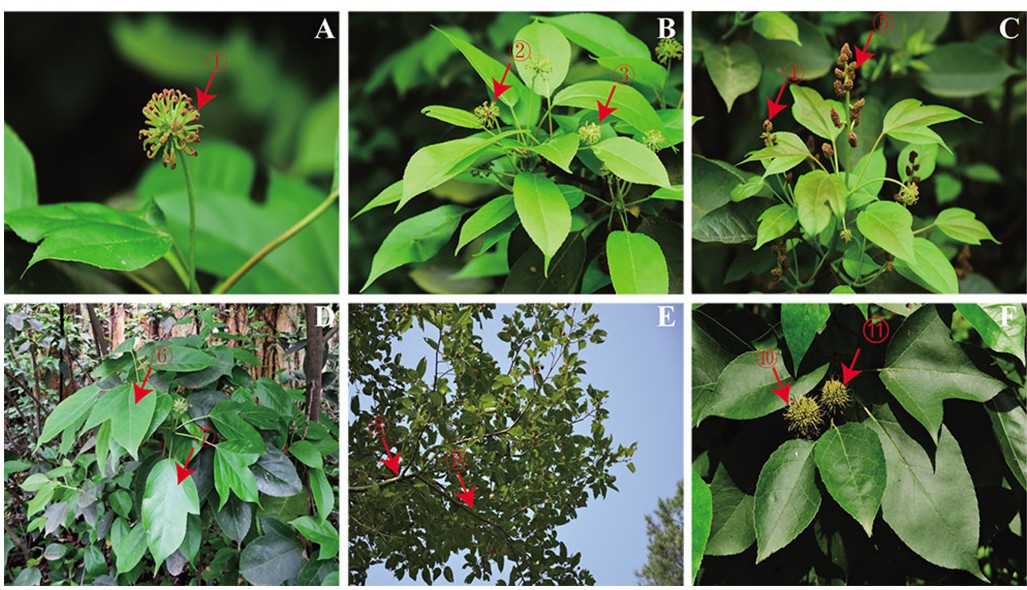

**Figure 1** (A–F) **Various tissues of *Semiliquidambar cathayensis*.** Female flower:1–3. Male flower:4–5. Leaf:6–7. Branch:8–9. Fruit: 10–11.

Burlington, MA, USA) for filtration. The extract (five mL) was added into a 10 mL flask, and then 5% NaNO2 solution (0.3 mL) was added. The samples were mixed and placed for 6 min. After that, 5% Al (NO3)3 solution (0.3 mL) was added to the flask. The sample was fully mixed and placed for 6 min, 4% NaOH solution (4.4 mL) was added. The samples were mixed well and placed for 12 min. Thereafter, the absorbance of the samples were read at 510 nm and rutin was used as the standard for the calibration curve.

Total triterpenoid acid content was determined according to the method described by *Xia et al. (2016)* with some modifications. The powder (10 g) was placed in methanol (100 mL), boiled at 80 °C for 3 h. The solution was cooled and weighed again. Thereafter, the mass loss was compensated with methanol to obtain the sample solution. The sample solution (0.3 mL) was evaporated methanol in a water bath; then, 5% vanillin glacial acetic acid solution (0.3 mL) and perchloric acid (1.0 mL) were added and mixed well. The sample solution was developed in color in a 60 °C water bath for 45 min and cooled in an ice bath. After that, glacial acetic acid was added to a constant volume to 5.0 mL. The absorbance of the sample solution was read at 550 nm, and ursolic acid was used as the standard of the calibration curve.

Total polyphenol content was determined according to the method described by *Chen et al. (2013)* with some modifications. The powder (10 g) was added into a 100 mL conical flask, then, 80% ethanol (50 mL) was added into the flask, which was sealed for 3 days. The conical flask was placed in an ultrasonic cleaner for extraction in three times. The collected filtrate was concentrated on the rotary evaporator (40 °C) until alcohol-free. The alcohol-free extract was transferred into a 100 mL volumetric flask, and the volume was replenished with distilled water to 100 mL. The solution (one mL) was then added

into a 10 mL volumetric flask, and the volume was replenished with distilled water to 10 mL to prepare the sample solution. The sample solution (1.0 mL) was added into a 10 mL volumetric flask, and then distilled water (three mL) and FC chromogenic agent (0.5 mL) were added. After being shaken, 20% NaCO3 (1.5 mL) was added within 8 min. The volume was made up with distilled water, and the flask was placed in a 75 °C water bath for 10 min. Absorbance of the sample was read at 760 nm, and gallic acid was used as the standard for a calibration curve.

## Qualitative analysis of seven chemical compounds by high performance liquid chromatography (HPLC) assays

The contents of seven chemical compounds in different geographic origin and tissue type of *S. cathayensis* were quantitatively determined according to the method described by *Belboukhari et al., 2010*, *Hu et al., 2016*, *Li et al., 2002*, *Silva et al., 2002* and *Zu et al., 2006* with some modifications. The different geographic origin and tissue type of *S. cathayensis* were air-dried and ground in a mill, filtered through a 60 mesh sieve. The powdered samples (0.1 g) were mixed with one mL methanol, and then moved into EP tubes. The extract samples were obtained by ultrasonic extraction for 60 min. The supernatants were obtained through 0.45 μm PTFE filter (Waters, Milford, MA, USA) before injection (10 mL) into the HPLC-DAD system (Rigol, Suzhou, China) . HPLC was carried out using the reversed-phase C18 column (Rigol, 250 × 4.6 mm 2 i.d., 5 μm) at room temperature. The mobile phase, wavelength of detection and retention time were shown in Data S1. The system consisted of the analytical HPLC unit in conjunction with a column-heating device set at 30 °C. Elution was carried out isocratically, at a solvent flow rate of 0.8 mL/min. The injection volume was 10 μL. The quantification of seven chemical compounds were achieved by the absorbance recorded in the chromatogram relative to the external standard, using the default baseline construction technique to integrate the peaks in the chromatogram (Fig. 2).

## Qualitative analysis of oleanolic aid and ursolic acid by HPLC-MS

Ursolic acid and oleanolic acid are triterpenoid isomers with exactly the same chemical structures, and the only difference between them is the position of a methyl group (*Tian et al., 2010*); moreover, there is only one conjugated double bond, which leads to weak UV absorption, low sensitivity and unsatisfactory baseline results when determining their concentration by HPLC. In contrast, mass spectrometry (MS) could be used as a detector to directly monitor the molecular ion peak, which eliminates the interference of other substances and obtain relatively high sensitivity. Therefore, the content of oleanolic acid and ursolic acid were determined according to the method described by *Zhao et al. (2015)* with some modifications.

The different geographic origin and tissue type of *S. cathayensis* were air-dried and ground in a mill, and then sieved through a 60 mesh sieve. Each sample was weighed about 0.6 g accurately and placed into a capped 100 mL conical flask. Then the sample was extracted with 60 mL of 90% ethanol in ultrasonic bath for 1 h at room temperature. After that, the solution was filtered and dried, and the residue was reconstituted in a five mL

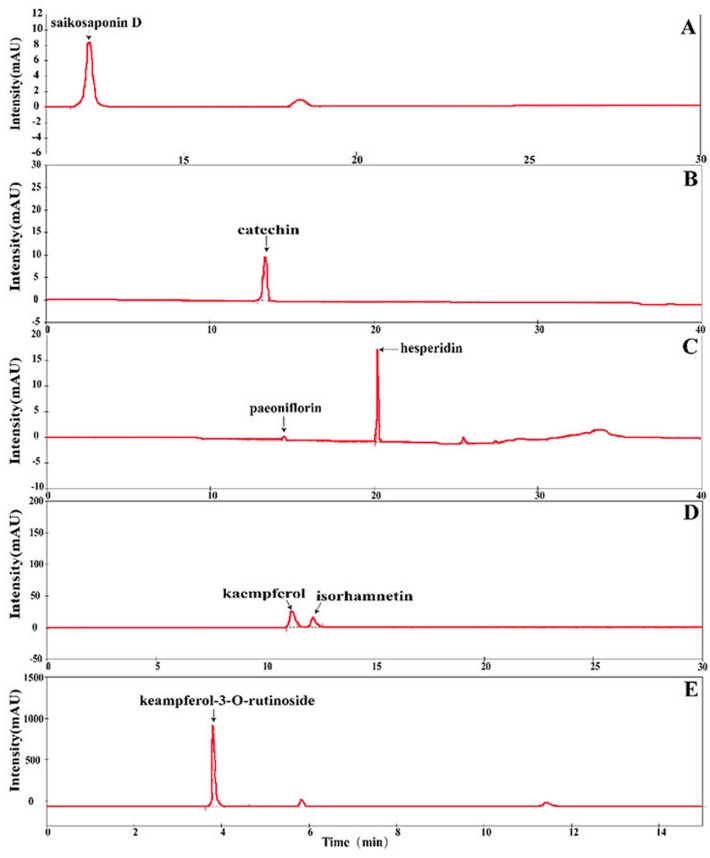

**Figure 2 Separation of chromatograms of various compounds with the HPLC method.** (A) Saikosaponin D. (B) Catechin. (C) Paeoniflorin and hesperidin. (D) Kaempferol and isorhamnetin. (E) Keampferol-3-O-rutinoside.

volumetric flask with methanol, and then pass through a 0.45 μM membrane filter. At the end, the sample solution was injected into the HPLC system with the injection volume 10 μL.

The HPLC-MS system consisted of an Agilent 1260 HPLC system and an Agilent 6420A MS system (Agilent Technologies, Santa Clara, CA, USA). The chromatographic separation was realized on an Agilent Poroshell 120 SB-C18 reversed phase chromatographic column (2.1 × 150 mm, 2.7 μm; Agilent Technologies) at 30 °C column temperature. The flow rate was constant at 0.3 mL/min. Eluent A was 5% acetonitrile/water (0.1% formic acid) and B was acetonitrile (0.1% formic acid)(HPLC-MS grade; TCI, Tokyo, Japan). The optimal MS conditions are shown in Data S2. Ursolic acid and oleanolic acid (10 mg) were weighed respectively, and added ethyl acetate (two mL) (HPLC-MS grade; TCI, Tokyo, Japan). A certain amount of the same solvent was added for further dilution to obtain the 20, 100, 500, 1,000, 2,500 and 5,000 ppm working solutions. The calibration curve was obtained by plotting the ratio of peak areas of ursolic acid and oleanolic acid to the concentration of the sample to quantify the concentration of the oleanolic acid and ursolic acid in the sample (Fig. 3).

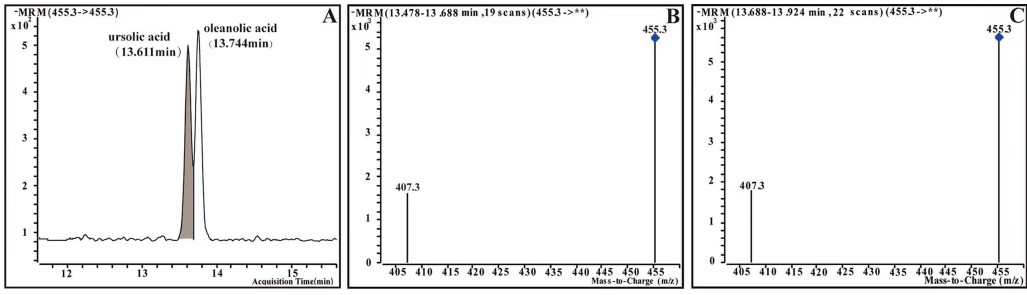

**Figure 3 Multiple-reaction monitoring (MRM) chromatograms and mass spectrum of ursolic acid and oleanolic acid of standards.** (A) MRM chromatograms of standards. (B) Mass spectrum of ursolic acid (UA). (C) Mass spectrum of oleanolic acid (OA).

## Data analysis

To compare the differences of different medicinal components among different samples, we first calculated the mean and variance of the three detected values of each sample, and compared the differences of the concentration of the same medicinal components in different samples by variance significance analysis. Tukey's HSD test was used to compare the contents of the different medicinal components extracted from the sampled plant tissues.

Correlations among the contents of the different medicinal components and different factors were analyzed by calculating Pearson product–moment correlation coefficients, and the medicinal components that were most strongly related to different factors were identified (*Xu et al., 2021*).

To determine the relationship between geographical origin and tissue group, the plant samples were clustered after determining the contents of different medicinal components. The geographical origin and tissue groups were classified using the inter-group connection method (*Qian et al., 2020*). All statistical analyses of the data were performed using SPSS, version 26 (IBM Corp. Armonk, NY, USA), with statistical significance set at $p < 0.05$.

## RESULTS

### Chemical composition of *S. cathayensis* plants from different geographical origins

There are significant differences in the contents of total flavonoid, total triterpenoid acid and total polyphenol in the leaves of plants from different geographical origins (Fig. 4, $p < 0.05$). For example, the contents of total flavonoids in the leaves of Jingzhou county plants were the highest, and the contents of total triterpenoid acid and polyphenol in Shangsi county were the highest. The contents of the three substances in the leaves of plants of any geographical origin are polyphenols > triterpenoid acid > flavonoids. In addition to these three substances, the nine substances that play a major medicinal role, except catechin and hesperidin, also had significant differences in the leaves of plants of different geographical origins (Table 2, $p < 0.05$), such as the highest concentration of saikosaponin D in the leaves of Dayu county, the highest concentration of paeoniflorin and

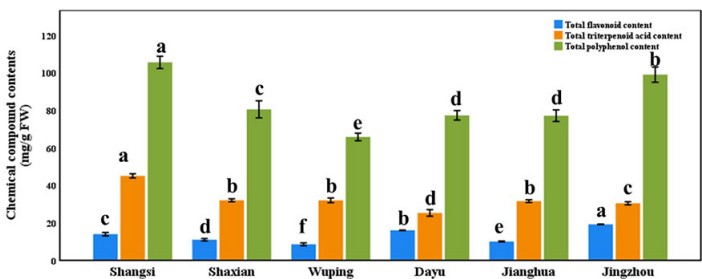

**Figure 4  Total flavonoid, total triterpenoidic acid, and total polyphenol contents in different geographical origin *Semiliquidambar cathayensis*.** Bars indicate standard error (±), with different lowercase letters indicating that different treatments of the same pollen source were significantly different from each other according to Tukey's test (*$P < 0.05$).

keampferol-3-O-rutinoside in Jingzhou county, the highest concentration of kaempferol and isorhamnetin in Jianghua, and the highest concentration of ursolic acid in Shaxian county. The concentration of oleanolic acid in Shangsi county was the highest.

From the results of analysis of variance alone, it is impossible to judge whether there is a correlation between the concentration of medicinal components and geographical origin, so the correlation between the chemical composition contents and geographical origin was analyzed (Table 3, $p < 0.01$). The results show that not all substances were related to geographical origin, and even in substances with significant differences unrelated to geographical origin, such as total flavonoid, total polyphenol, saikosaponin D, kaempferol and isorhamnetin, their correlation coefficients were low. However, it is worth noting that, although the concentration of catechin and hesperidin in the sample had no significant difference among multiple geographical origins, it had a high correlation with the geographical origin, and the correlation coefficients were 0.63 and 0.72, respectively.

In order to further understand the similarity of the composition of medicinal compounds of different geographical origin, the concentration data of medicinal components significantly related to geographical origins were used to cluster different geographical origins (Fig. 5). From the results of clustering, the six geographical origins can be grouped into three groups; that is, Shangsi county and Shaxian county are grouped into one group (group 1), Wuping county and Dayu county are clustered into one group (group 2), Jianghua county and Jingzhou county are clustered into one group (group 3). From the analysis of the earliest branch nodes, the relationship between Shangsi, Shaxian, Wuping and Dayu county are relatively close, while Jianghua county and Jingzhou county are obviously separated from these four counties. From the point of view of the nodes of the last branch, the similarity between the two regions of group 1 is the highest, the similarity between group 2 is lower than that between group 1, and the two areas of group 3 are the least similar.

## Chemical compounds of different tissues of *S. cathayensis*

There were significant differences in the concentrations of total flavonoid, total triterpenoid acid and total polyphenol in different tissues of the same plant (Fig. 6, $p < 0.05$). The highest

Tian et al. (2023), *PeerJ*, DOI 10.7717/peerj.15484

**Table 2  Chemical compound contents (μg/g) tested by HPLC and HPLC-MS in different geographical origin of *Semiliquidambar cathayensis*.**

| | SaikosaponinD | Catechin | Paeoniflorin | Hesperidin | Kaempferol | Isorhamnetin | Keampferol-3-O-rutinoside | Ursolic acid | Oleanolic acid |
|---|---|---|---|---|---|---|---|---|---|
| Shangsi | 63.80 ± 0.45f | 293.22 ± 0.96a | 31.52 ± 0.47f | 1.63 ± 0.03a | 208.88 ± 0.76f | 2.10 ± 0.05f | 976.44 ± 0.46e | 4.36 ± 0.23c | 4.79 ± 0.11a |
| Shaxian | 74.07 ± 0.12c | 569.75 ± 0.38a | 142.03 ± 0.46e | 24.90 ± 0.02a | 1718.13 ± 0.81b | 67.58 ± 0.28c | 897.48 ± 0.34f | 8.61 ± 0.16a | 4.40 ± 0.09b |
| Wuping | 79.07 ± 0.71b | 1,998.25 ± 0.74a | 158.95 ± 0.19d | 43.70 ± 0.04a | 313.82 ± 0.47e | 151.01 ± 0.04b | 1,842.26 ± 0.33d | 5.44 ± 0.09b | 3.19 ± 0.12c |
| Dayu | 91.22 ± 0.26a | 932.09 ± 0.68a | 682.97 ± 0.18c | 10.28 ± 0.04a | 1482.05 ± 0.42c | 64.83 ± 0.32d | 1,881.13 ± 0.47c | 0.43 ± 0.06f | 0.3 ± 0.11f |
| Jianghua | 65.12 ± 0.11e | 1,997.76 ± 0.31a | 1,054.05 ± 0.37b | 83.21 ± 0.41a | 2,485.09 ± 1.36a | 288.10 ± 0.13a | 2,372.12 ± 42.56b | 1.12 ± 0.08e | 0.92 ± 0.03d |
| Jingzhou | 72.08 ± 0.19d | 828.24 ± 0.25a | 1,649.65 ± 0.63a | 342.79 ± 0.45a | 1,303.64 ± 0.34d | 64.17 ± 0.33e | 2,676.75 ± 0.49a | 1.4 ± 0.03d | 0.83 ± 0.02e |

**Notes.**

Within columns, different lowercase letters indicate significant differences at $P < 0.05$.

Tian et al. (2023), *PeerJ*, DOI 10.7717/peerj.15484

**Table 3  Pearson product–moment correlation coefficients of geographical origin with different chemical compound contents.**

| Factor | Total flavonoid content | Total triterpenoid content | Total polyphenol content | Saikosaponin D | Catechin | Paeoniflorin | Hesperidin | Kaempferol | Isorhamnetin | Keampferol-3-O-rutinoside | Ursolic acid | Oleanolic acid |
|---|---|---|---|---|---|---|---|---|---|---|---|---|
| Site | 0.31 | −0.73[**] | −0.23 | 0.14 | 0.63[**] | 0.99[**] | 0.72[**] | 0.42 | 0.27 | 0.93[**] | −0.59[**] | −0.83[**] |

**Notes.**
[**]T-test for all variables, $P < 0.01$.

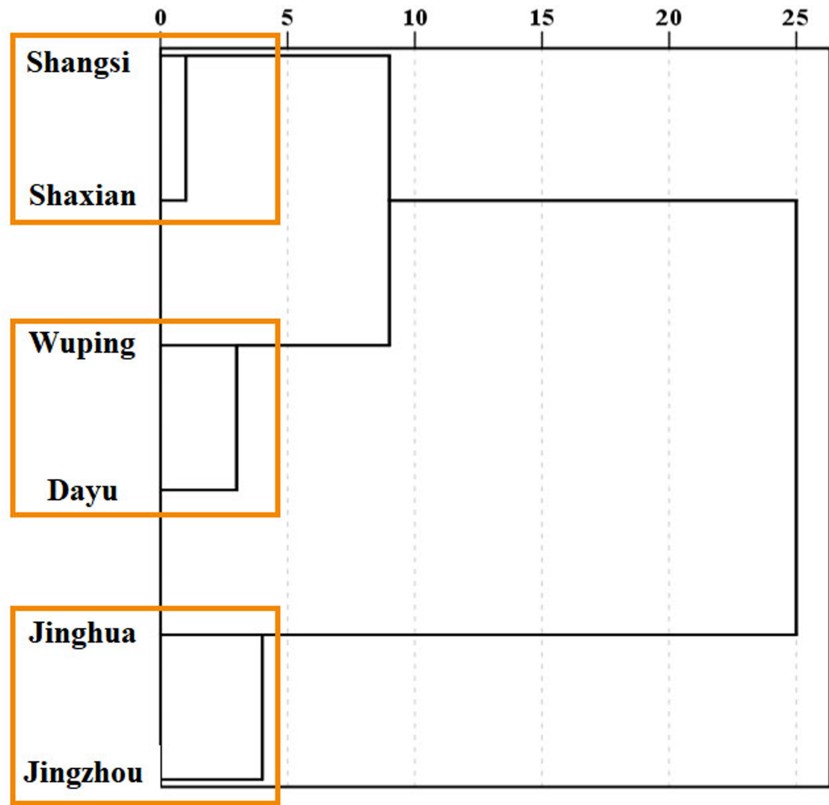

**Figure 5 Clustering of *Semiliquidambar cathayensis*.** Total triterpenoidic acid, catechin, paeoniflorin, hesperidin, keampferol-3-O-rutinoside, ursolic acid and oleanolic acid contents with chemical compounds related of geographical origin. The geographical origin and tissue groups were classified using the inter-group connection method.

concentration of total flavonoid in leaves, the highest concentration of total triterpenoid acid in female flowers and the highest concentration of total polyphenol in male flowers. The contents of three kinds of substances in fruit, root and stem were significantly lower than those in leaves, male flowers and female flowers, and they contained almost no flavonoids. Among the other nine main medicinal substances (except for catechin), there were significant differences among different tissues (Table 4, $p < 0.05$). Paeoniflorin, hesperidin, kaempferol and Keampferol-3-O-rutinoside; although the contents of total flavonoid, total triterpenoid acid and total polyphenol in roots were low, the concentrations of oleanolic acid and ursolic acid were significantly higher than those in other tissues; in addition, the concentration of saikosaponin D in stem was the highest.

The correlation between different medicinal compounds and tissues is different (Table 5, $p < 0.01$). The results of correlation analysis showed that most of the indexes such as total flavonoid, total triterpenoid acid and total polyphenol, catechin, paeoniflorin, hesperidin, kaempferol, keampferol-3-O-rutinoside and oleanolic acid were significantly correlated with plant tissues, only saikosaponin D, isorhamnetin and ursolic acid were not significantly
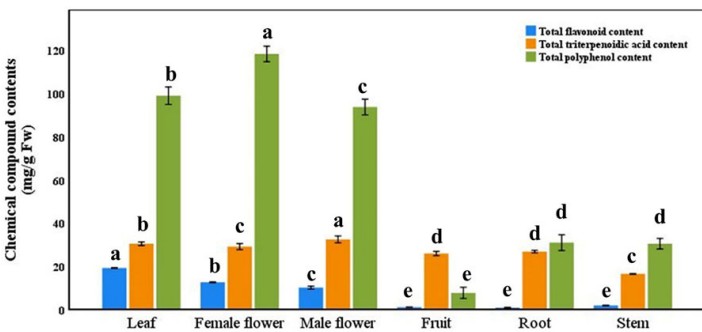

**Figure 6 Total flavonoid, total triterpenoidic acid, and total polyphenol contents in different tissues *Semiliquidambar cathayensis*.** Bars indicate standard error (±), with different lowercase letters indicating that different treatments of the same pollen source were significantly different from each other according to Tukey's test (*$P < 0.05$).

correlated with plant tissues, and their correlation coefficients were 0.42, 0.37 and 0.44, respectively.

Using the medicinal compounds which are significantly related to plant tissues, different plant tissues were analyzed by cluster analysis (Fig. 7). The results of cluster analysis show that there is obvious difference between tissues, and there is only one branch in the cluster map. Six tissues can be divided into two categories. Roots, stems, fruits, female flowers and male flowers are clustered into one group, and leaves are divided into separate groups.

## DISCUSSION

While analyzing the leaves of *S. cathayensis* specimens from six different regions, we observed that the amount and type of medicinal components present was dependent on the geographical origin of the samples (*Hwa & Ju, 2022*; *Popović et al., 2021*; *Vaneková et al., 2020*), although not all medicinal ingredients changed with geographic origin. Nevertheless, total leaf triterpenoids, catechin, paeoniflorin, hesperidin, keampferol-3-O-rutinoside, and oleanolic acid contents may be used as biomarkers to determine the geographical origin of *S. cathayensis*. Hence, our findings directly demonstrate that geographical location directly affect plant medicinal value. (*Wen et al., 2020*; *Zutic et al., 2016*). It is noteworthy that no obvious correlation was detected between latitude and plant chemical composition, suggesting that many origin-related factors other than latitude may contribute to the observed differences on medicinal component contents. In comparison with *Ye et al. (2021a)*, we find that the clustering results based on metabolites were similar to those based on genes; for example, Jianghua county is independent of Wuping county, Shangsi county, Dayu county and Shaxian county, which may indicate that the concentration of metabolites of *S. cathayensis* plants in Hunan Province is quite different from that in other areas, not only due to the difference of geographical environment. It is also closely related to genetic differences. Although Shaxian county and Wuping county are located in the same province, they are grouped into the same category, which is consistent with the results of previous studies, which shows that there are great differences in the metabolites

**Table 4  Chemical compound contents (μg/g) tested by HPLC and HPLC-MS in different tissues of *Semiliquidambar cathayensis*.**

| Tissues | SaikosaponinD | Catechin | Paeoniflorin | Hesperidin | Kaempferol | Isorhamnetin | Keampferol-3-O-rutinoside | Ursolic acid | Oleanolic acid |
|---|---|---|---|---|---|---|---|---|---|
| Leaf | 72.08 ± 0.19b | 828.24 ± 0.25a | 1,649.65 ± 0.63a | 342.79 ± 0.45a | 1,303.64 ± 0.34a | 64.17 ± 0.33b | 2,676.75 ± 0.49a | 1.4 ± 0.03f | 0.83 ± 0.02e |
| Female flower | 15.43 ± 0.14f | 194.12 ± 0.4a | 79.57 ± 0.25c | 159.19 ± 0.28b | 236.55 ± 0.48b | 5.36 ± 0.20d | 303.11 ± 0.70c | 4.18 ± 0.09c | 1.22 ± 0.06d |
| Male flower | 27.59 ± 0.21e | 539.15 ± 0.47a | 81.1 ± 0.69b | 61.04 ± 0.15c | 59.78 ± 0.52c | 73.71 ± 0.51a | 1,150.98 ± 0.58b | 1.32 ± 0.04e | 0.32 ± 0.03f |
| Fruit | 33.07 ± 0.36d | 67.7 ± 0.32a | 20.96 ± 0.63d | 13.07 ± 0.05f | 2.39 ± 0.29e | 1.79 ± 0.12e | 19.29 ± 0.25e | 4.81 ± 0.05b | 2.52 ± 0.03b |
| Root | 54.33 ± 0.35c | 294.73 ± 0.26a | 5.47 ± 0.09e | 29.5 ± 0.36e | 2.57 ± 0.24e | 2.07 ± 0.04e | 16.61 ± 0.39f | 5.12 ± 0.07a | 3.17 ± 0.02a |
| Stem | 90.85 ± 0.17a | 21.41 ± 0.28a | 10.68 ± 0.25f | 44.75 ± 0.05d | 26.86 ± 0.56d | 8.86 ± 0.13c | 215.75 ± 0.68d | 1.91 ± 0.05d | 1.38 ± 0.07c |

**Notes.**

Within columns, different lowercase letters indicate significant differences at $P < 0.05$.

Tian et al. (2023), *PeerJ*, DOI 10.7717/peerj.15484

**Table 5  Pearson product–moment correlation coefficients of tissues with different chemical compound contents.**

| Factor | Total flavonoid content | Total triterpenoid content | Total polyphenol content | Saikosaponin D | Catechin | Paeoniflorin | Hesperidin | Kaempferol | Isorhamnetin | Keampferol-3-O-rutinoside | Ursolic acid | Oleanolic acid |
|---|---|---|---|---|---|---|---|---|---|---|---|---|
| Tissues | −0.82** | −0.76** | −0.74** | 0.42 | −0.71** | −0.88** | −0.76** | −0.78** | −0.37 | −0.76** | 0.44 | 0.62** |

**Notes.**
**T-test for all variables, $P < 0.01$.

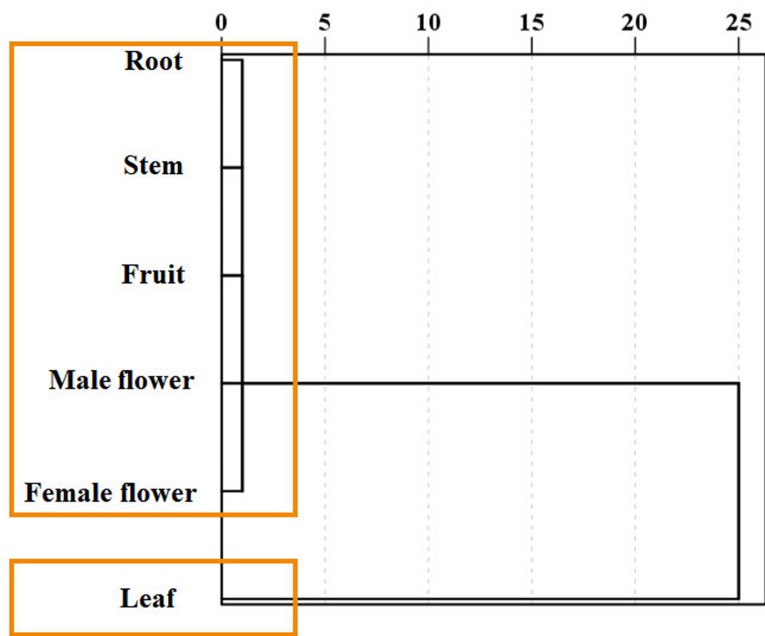

**Figure 7 Clustering of *Semiliquidambar cathayensis*.** Total flavonoid, total triterpenoidic acid, total flavonoid, catechin, paeoniflorin, hesperidin, kaempferol keampferol-3-O-rutinoside, oleanolic acid contents that chemical compounds related. The geographical origin and tissue groups were classified using the inter-group connection method.

of plants of geographical origin with little difference in geographical environment in the same province (*Ye et al., 2021b*).

Triterpenoid acids are mainly used in medicine owing their wide range of biological activities. This is particularly true for some pentacyclic triterpenes with unique structures, such as oleanolic and ursolic acids. Oleanolic acid is often used as a drug to protect the liver, or added to products as an anti-inflammatory and antioxidant ingredient (*Luo et al., 2018*). In addition, oleanolic acid reportedly shows anticancer effects, thereby holding high medicinal value and potential (*Furtado et al., 2008*; *Şenol et al., 2022*; *Zeng et al., 2022*). Ursolic acid, similar to oleanolic acid, can prevent cancer, but its mechanisms of action are different; for example, ursolic acid can protect against DNA damage, while it also enhances DNA repair (*Furtado et al., 2008*; *Han et al., 2022*; *Xiaoxia et al., 2022*; *Zeng et al., 2022*). The contents of these two important substances in *S. cathayensis* are significantly related to plant geographical origin, but further studies are necessary to identify which factors may contribute for this, such as local soil conditions, microclimate, or other factors, as longitude and latitude do not seem to be relevant to any large extent. Based on our data, we speculate that *S. cathayensis* from Jingzhou county has the highest medicinal value, because the contents of many medicinal ingredients, including catechin, hesperidin, and paeoniflorin, are significantly higher in plants from this region than in those from other geographical locations.

Consistent with public perception, we confirmed that *S. cathayensis* leaves are an excellent medicinal material (*Liu et al., 2022b*). However, we also found that the amount and type of medicinal compounds differ significantly among the tissues/organs of the plants collected from different locations. Hence, a better understanding of the medicinal value of the different tissues of *S. cathayensis* is necessary to ensure that the appropriate parts are selected according to the desired outcome (*Kirakosyan et al., 2003*; *Ma et al., 2022*; *Politi et al., 2004*; *Tan et al., 2007*). Each medicinal plant contains polyphenol, triterpenoid acid, and flavonoids, and other secondary metabolites that are specifically synthesized in different tissues and are then transported to other parts of the plant through specific transport routes, where they accumulate (*Li & Wu, 2018*). Therefore, the contents of medicinal compounds in different tissues of plants of the same geographical origin can greatly differ, which further determines that not every plant organ or tissue can be used as medicine. For example, the content of medicinal compounds in *Magnolia officinalis* varies with the height of the bark (*Ma et al., 2022*). Similarly, the medicinal value of the main aboveground parts of *Lagotis integra* are higher than that of the underground parts (*Gao et al., 2022*). Notably, in *S. cathayensis*, not every medicinal ingredient was found to be significantly correlated with the different tissues; indeed, saikosaponin D and isorhamnetin levels were similar across all tissues analyzed. Overall, most medicinal ingredients were detected at higher levels in the leaves of *S. cathayensis*, which further explains their widespread use in traditional Chinese medicine. In contrast, the contents of oleanolic and ursolic acids were lower in the leaves, as compared with the other tissues, whereas they were highest in the roots. These findings suggest that the roots may be more suitable for extracting oleanolic and ursolic acids, whereas the leaves are more suitable for extracting other medicinal compounds.

## CONCLUSIONS

In this study, we used colorimetric and chromatographic methods to determine the levels of several chemical compounds of *S. cathayensis* with medicinal potential with regard to the geographical origin of the plant and the different plant tissues/organs. Our analysis confirmed that total flavonoid, total triterpenoid acid, and total polyphenol contents accumulation in *S. cathayensis* leaves were dependent, to some extent, on the geographical origin of the plant. Further, the concentration of medicinal chemicals varied significantly across plant tissues; for example, catechins and other substances mainly accumulated in leaves, whereas oleanolic and ursolic acids mainly accumulated in the roots. Therefore, the geographical origin and tissue type are all factors that impact on the concentration of medicinal compounds in *S. cathayensis*, and should therefore be considered when selecting *S. cathayensis* plants for the production of medicinal treatments. These results show that the comprehensive medicinal value of the leaves of *S. cathayensis* in Jingzhou county is the highest, but the root should be selected first to collect oleanolic acid and ursolic acid.

## ACKNOWLEDGEMENTS

We thank Editage for English language editing.

### Funding

This research was funded by the Changsha Natural Science Foundation (Grant number kq2202356), the Natural Science Foundation of Hunan Province (2023JJ60116), the Natural Science Foundation of Changsha City (Project No.: kq2202356), and the Hunan Forestry Science and Technology Innovation Plan Project (Grant number XLK202106-2). The funders had no role in study design, data collection and analysis, decision to publish, or preparation of the manuscript.

### Grant Disclosures

The following grant information was disclosed by the authors:
Changsha Natural Science Foundation: kq2202356.
Natural Science Foundation of Hunan Province: 2023JJ60116.
Natural Science Foundation of Changsha City: Project No.: kq2202356.
Hunan Provincial Forestry Science and Technology Innovation Program: Project No.: XLK202106-2.

### Competing Interests

The authors declare there are no competing interests.

### Author Contributions

- Xiaoming Tian conceived and designed the experiments, performed the experiments, analyzed the data, prepared figures and/or tables, authored or reviewed drafts of the article, contributed reagents, materials, analysis tools, and approved the final draft.
- Hao Lv performed the experiments, authored or reviewed drafts of the article, and approved the final draft.
- Guangfeng Xiang performed the experiments, prepared figures and/or tables, and approved the final draft.
- Jing Peng performed the experiments, analyzed the data, authored or reviewed drafts of the article, and approved the final draft.
- Gaofei Li performed the experiments, analyzed the data, prepared figures and/or tables, and approved the final draft.
- Yan He performed the experiments, analyzed the data, prepared figures and/or tables, authored or reviewed drafts of the article, and approved the final draft.
- Fan Zhang performed the experiments, prepared figures and/or tables, and approved the final draft.
- Cun Mou conceived and designed the experiments, performed the experiments, analyzed the data, prepared figures and/or tables, authored or reviewed drafts of the article, and approved the final draft.

## Data Availability

The raw measurements are available in the Supplementary Files.

## Supplemental Information

Supplemental information for this article can be found online at http://dx.doi.org/10.7717/peerj.15484#supplemental-information.

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
