# Peer review of "Influence of geographic origin and tissue type on the medicinal chemical compounds of Semiliquidambar cathayensis"

_PeerJ, doi:10.7717/peerj.15484_

## Round 0.1 · original submission · Major Revisions

I agreed with the reviewers, the materials and methods part are not clear, make it difficult for audiences to understand this study from the beginning. Although 2 of the 3 reviewers recommended rejecting this manuscript, I would like to give the authors one chance to improve this manuscript. Please make substantial revisions to this manuscript based on all the review comments and give point-to-point explanations with possible logical reasons.

Besides responses to each reviewer,

1. Please supplement references for Spectrophotometric analysis, HPLC, HPLC-MS, as well as for the plant sample preparation method.

2. Based on the results, authors compared the samples from different geographical origins, tissues, old and young leaves, which should be firstly introduced in the materials and methods as experimental design. Please make supplement about the detailed origin information (I recommend authors make a table), definition/selection standard of the ‘old leaves’ and ‘young leaves’.

3. For flower and fruit samples, were the whole flower/fruit, including the petal, ovary, very hard spiky pericarp, and seeds included?

4. Please re-design the table format to make it easier to read.

5. Please add interpretation/description about the correlation analyses in results.

Reviewer 1 ·

Basic reporting

In this paper, colorimetric and chromatographic methods were used to study the influence of geographic origin and tissue type on the compounds of S. cathayensis, and the amount of paeoniflorin and other components can be regarded as biomarkers of geographical origin and tissue type. While the results of this article are interesting, there are some problems as follows. First, a total of two acids, three acid water concentrations, and five mobile phase gradients were used to determine the content of seven compounds, which was too complex, and the validation of the quantitative method is missing. Secondly, the number of geographic origins is not sufficient, and each origin is only one batch of samples. Can it represent the quality of the whole geographic origin? Thirdly, comparing the amount of target compounds in different tissues of S. cathayensis does not mention where it originated, and the distinction criteria for old and young leaves need to be mentioned in the main text. Finally, the chromatograms of the samples are not shown in the manuscript, and the chromatograms in the manuscript are not clear enough. Thus, I think this paper is not suitable for publication in PeerJ – The Journal of Life and Environmental Sciences at the current state, and it is necessary to make further supplements and improvements.

Experimental design

As mentioned above.

Validity of the findings

As mentioned above.

Additional comments

no comments

Reviewer 2 ·

Basic reporting

Dear Editor,

I attach main remarks to the article: Influence of geographic origin and tissue type on the medicinal chemical components of Semiliquidambar cathayensis.
Main remarks
• Not adequate statistical analysis makes it impossible to review the article.
• The lack of appropriate statistical analysis makes it impossible to review the article – an analysis of variance (there are more than 2 groups) should be a first step.
• The form of tables makes it impossible to review the article.
• Lack of characteristics of research sites (natural sites): vegetation, vertical structure of vegetation, competition, light conditions, temperature, water conditions, soil conditions – these factors determine the habitat conditions and secondary metabolites production/quality, not latitude. Additionally, altitude from 421m to 835m makes a difference.

Conclusion
This manuscript does not meet the basic requirements and is not suitable for publication in this form.

Experimental design

-

Validity of the findings

-

Additional comments

-

Reviewer 3 ·

Basic reporting

The manuscript is fairly well-written and easy to understand. The Introduction section provides sufficient background information to explain the rationale behind this study.
Suggestions for improvement:
1. Please provide sufficient information in the figure legend and footnotes of tables.
2. Images of higher quality should be used, for instance, Fig 2 is too blurry.
3. Many parts are not sufficiently described and discussed, for instance, the clustering of the samples.
4. "Young and old leaves" is abit subjective and having two samples does not really represent "different developmental stages"

Experimental design

Standard procedures were used but there is a lack of details in some aspects:
1. Please name the botanist who authenticate the plant samples and indicate if any specimens have been deposited in herbarium.
2. Please explain the criterion used to classify the samples into young and old leaves.
3. Please clarify the number of replicates or independent samples used in this study.
4. Please explain the basis for choosing the six locations.

Validity of the findings

Data analysis seems acceptable but could be improved.
1. The authors should discuss the insights from various comparison done in this study: 1. geographical location, 2. types of tissues, and 3. young and old leaves. Are their findings consistent with previous studies with the same objectives but on different plant species?
2. I would like to suggest to authors to rephrase the term "developmental stages" as only two samples were used.
3. Please include the relevant chromatogram and MS spectra of the extracts.

Additional comments

This study looks at the medicinal plant Semiliquidambar cathayensis from an interesting point of view but the results were not thoroughly or critically discussed to gain insights into the variation of chemical constituents in different settings such as geographical locations, types of tissues and young and old leaves.
Please consider to provide a more concise conclusion that sums up the findings of this study.

---

## Round 0.2 · Minor Revisions

Please revise the manuscript according to comments made by reviewers. If some comments can't be replied, authors should respond with your opinion and references. For example, it is possible to analyze 7 compounds through 1 HPLC method. However, authors should make sure the cited reference is valid and the method is thoroughly described either in this manuscript or in cited references. Authors should include HPLC chromatograms not only from standards but also from samples. Our objective is to make the study replicable, which will help the future researchers.

Reviewer 1 ·

Basic reporting

I'm very sorry for replying to you so late! I have read the author's revised manuscript and rebuttal letter, but the response to comments is not comprehensive. As for the complex analysis methods, I do not agree with the author's statement. Even though it is impossible to complete the determination of seven compounds through one HPLC method, it is completely feasible to use two or three methods for quantitative analysis. Unfortunately, this result seems to be unchangeable. Meanwhile, the reason for not conducting method verification is not explained. Therefore, I think the quantitative analysis part is not systematic and reliable. In addition, the spectrogram figures resupplied by the authors are still unclear, and is the “Chemical compounds of S. cathayensis” in the title superfluou? Therefore, the author's revision is not comprehensive and does not meet the requirements of publication. If the PeerJ – The Journal of Life and Environmental Sciences is to consider publishing this article, it would needs a major revision.

Experimental design

no

Validity of the findings

no

Additional comments

The text paragraph format, Tables and Figures need to be adjusted and beautified.

Reviewer 2 ·

Basic reporting

Dear Editor,

I attach main remarks to the article: Influence of geographic origin and tissue type on the medicinal chemical components of Semiliquidambar cathayensis.
Main remarks
• Not adequate statistical analysis makes it impossible to review the article.
• The lack of appropriate statistical analysis makes it impossible to review the article – an analysis of variance (there are more than 2 groups) should be a first step.
• The form of tables makes it impossible to review the article.
• Lack of characteristics of research sites (natural sites): vegetation, vertical structure of vegetation, competition, light conditions, temperature, water conditions, soil conditions – these factors determine the habitat conditions and secondary metabolites production/quality, not latitude. Additionally, altitude from 421m to 835m makes a difference.

Conclusion
This manuscript does not meet the basic requirements and is not suitable for publication in this form.

Experimental design

.

Validity of the findings

.

Additional comments

.

Reviewer 3 ·

Basic reporting

The authors have responded to my queries in the previous review. I do not have other technical comments, however, I would like to suggest language correction for the manuscript. "The total "triterpenoidi acid assay" should be "total triterpenoid content".

Experimental design

No comment

Validity of the findings

No comment

---

## Round 0.3 · Minor Revisions

Thanks for authors' carefully revision according to reviewers' comments. However, the HPLC chromatograms of the sample should be presented similar to Figure 2 instead of including the Agilent reports without detailed descriptions. I can not understand which page in that supplementary file is from which sample. Authors don't have to present chromatogram of all the samples, just present a representative one to prove that the HPLC method used in this study is valid. After this small revision, I will send this manuscript to reviewers to re-review it. Thank you. I am looking forward to your revision.

---

## Round 0.4 · Minor Revisions

I am grateful for the revisions authors have made on this manuscript. I have a few suggestions that I think would improve the readability of this manuscript.

First, please re-arrange the tables to make them easier to read. Currently, some of the tables have multiple lines in each cell. This can make it difficult to see the data in the table. I would suggest using a different format for the tables or you can change some of the pages to horizontal to have more space to accommodate wide tables. In table 3. The red-colored content should be revised.

---

## Round 0.5 · Major Revisions

The conclusion that the plants from Jingzhou county have the highest medicinal value does not seem to be justified. From Table 2,Ursolic acid and Oleanolic acid are LOWER in these plants, directly contradicting the text of lines 316-318.

Figure 7 legend truncated. Also weird spacing "k aempferol" instead of "kaempferol" etc.

line 344: remove "comprehensive" this is not a comprehensive analysis.

Each table seems to be presented twice.

---

## Round 0.6 · accepted · Accept

Thanks for the revision. I am glad to inform the authors that the manuscript has been accepted for publishment.